# Time-Series Sensory Analysis Provided Important TI Parameters for Masking the Beany Flavor of Soymilk

**DOI:** 10.3390/foods12142752

**Published:** 2023-07-19

**Authors:** Miyu Masuda, Yuko Terada, Ryoki Tsuji, Shogo Nakano, Keisuke Ito

**Affiliations:** Department of Food and Nutritional Sciences, Graduate School of Integrated Pharmaceutical and Nutritional Sciences, University of Shizuoka, 52-1 Yada, Suruga-ku, Shizuoka 422-8526, Japan; s22209@u-shizuoka-ken.ac.jp (M.M.); yukoterada@u-shizuoka-ken.ac.jp (Y.T.); f20017@u-shizuoka-ken.ac.jp (R.T.); snakano@u-shizuoka-ken.ac.jp (S.N.)

**Keywords:** time-series sensory evaluation, time-intensity method, masking, soymilk, flavor materials, machine learning

## Abstract

The aim of this study is to provide a new perspective on the development of masking agents by examining the application of their time-series sensory profiles. The analysis of the relationship between 14 time-intensity (TI) parameters and the beany flavor masking ability of 100 flavoring materials indicate that the values of AreaInc, DurDec, and AreaDec, TI parameters related to the flavor release in the increasing and decreasing phases, were significantly higher in the top 10 masking score materials than in the bottom 10 materials. In addition to individual analysis, machine learning analysis, which can derive complex rules from large amounts of data, was performed. Machine learning-based principal component analysis and cluster analysis of the flavoring materials presented AreaInc and AreaDec as TI parameters contributing to the classification of flavor materials and their masking ability. AreaDec was suggested to be particularly important for the beany flavor masking in the two different analyses: an effective masking can be achieved by focusing on the TI profiles of flavor materials. This study proposed that time-series profiles, which are mainly used for the understanding of the sensory characteristics of foods, can be applied to the development of masking agents.

## 1. Introduction

Flavor, consisting of taste and aroma, is an important element of food that determines consumer preferences and purchase intentions. In the food industry, masking unpleasant flavors is a significant challenge. The addition of flavoring agents is one of the most commonly used approaches to masking unpleasant flavors in food [1]. For example, based on cooking experiences, flavoring agents related to spices and the Maillard reaction were added to soy-based meat to mask its beany flavor [2,3]. In other studies, flavoring materials that effectively mask the unpleasant odors of orange juice and Pu-erh tea were screened using gas chromatography-olfactometry [4,5]. As in the abovementioned examples, various attempts have been made to mask unpleasant flavors using flavoring agents. A detailed understanding of the sensory properties of the flavoring materials would lead to a more strategic masking of unpleasant flavors.

The characteristics of flavoring materials are typically analyzed using sensory evaluation methods [6,7,8]. Sensory evaluation methods are generally classified into two types: endpoint types and time-series types. The endpoint type is a method of statistically evaluating the sensory characteristics of food and is suitable for the comprehensive evaluation of flavor characteristics. Time-series type is a method of dynamically evaluating the sensory characteristics of food, which enables the analysis of the temporal changes in flavor that occur in the mouth [9]. The time-intensity (TI) method is one of the time-series sensory evaluation methods. It is a technique that analyzes the temporal changes in the intensity of one sensory attribute [10,11]. TI analysis yields dynamic parameters (time-series sensory profiles) (e.g., maximum intensity, time point of maximum intensity, and end point of sensation) [12,13]. Moreover, the TI method is used to obtain a detailed understanding of the temporal changes in flavor intensity that are difficult to capture with the endpoint method. For instance, the temporal changes in flavor intensity in fruit juices, wine, beer, and ice cream are analyzed using the TI method [14,15,16,17,18].

In this study, we aimed to provide a new perspective on the development of masking agents by assessing the application of time-series sensory profiles for analyzing the flavor materials. Soymilk was used as a model food with an unpleasant flavor because soy, which has a low environmental impact, is being anticipated as an alternative protein source [2,3]. TI profiles for the 100 flavoring materials were obtained, and then a detailed relationship analysis between these profiles and the beany flavor masking ability of the materials was conducted. As a result of the analysis, the TI parameters that are related to the masking ability were successfully identified. Time-series sensory profiles can be applied to the development of effective masking materials.

## 2. Experimental Section

### 2.1. Food-Flavoring Materials

One hundred commercially available food-flavoring materials (T&M Corporation, Chiba, Japan) were used for the sensory evaluation. These flavoring materials were classified into three types based on their chemical features: essence type (e), oil type (o), and flavor type (f) (Table 1). Each material name was denoted as flavor name_type (e, o, or f). For example, apple essence, cinnamon oil, and coffee flavor were denoted as apple_e, cinnamon_o, and coffee_f, respectively. The essence-type materials have water solubility and high volatility and are dissolved into alcohols. The oil-type materials are characterized by their oil solubility and low volatility and are dissolved in glycerol fatty acid esters and vegetable oils. The flavor-type materials have intermediate polarity and high volatility and are dissolved into propylene glycol and glycerin.

In this study, a total of 100 flavor materials were used, including 33 essence-type, 33 oil-type, and 34 flavor-type materials (Appendix A). The detailed flavor categories and numbers belonging to each category were as follows: 55 fruits (8 berries, 12 citrus, and 35 other flavors), 5 nuts, 7 herbs and spices, 3 flowers, 21 sweets, and 9 drink flavors. The flavor materials consisted of artificially blended materials and natural-derived materials, such as essential oils. Some of the materials have the same food name, such as orange essence and orange oil, but have different compositions.

### 2.2. Sample Preparation for Sensory Evaluation

In our preliminary study, the beany flavor of soymilk interfered with the TI analysis of the food-flavoring materials. To overcome this issue, a fake soymilk without beany flavor was prepared and used for the TI analysis. Considering that the fake soymilk used in the study was almost tasteless and odorless, its impact on the flavor TI evaluation would be minimal. This fake soymilk had the same content of carbohydrates, proteins, and fats as plain soymilk, but was prepared by mixing dextrin, soy protein RT-1, almond milk, and water to eliminate the beany flavor. Table 2 shows the composition of the fake soymilk. The almond milk was purchased from a local supermarket. Soy protein RT-1 and dextrin were provided by Fuji Oil Holdings, Inc. (Osaka, Japan). The food-flavoring materials were added to the fake soymilk at a concentration of 0.2% as per the manufacturer’s recommendation. In our pilot study, we determined the appropriate concentration to yield the TI curve for each of the flavoring materials. The fake soymilk (10 mL) and plain soymilk (10 mL) were each mixed with one of the food-flavoring materials (20 µL) and used for the TI analysis and the beany flavor masking test, respectively. The plain soymilk was purchased from a local supermarket.

### 2.3. Panelists and Training

The panelists were 10 adults (3 men and 7 women) between the ages of 21 and 26 (average age: 22.9 years) and were non-smokers. The outline of the research was explained to each panelist, and approval for data use was obtained from all panelists. The participants were asked to refrain from eating, brushing their teeth, or talking during and at least 1 h before testing. All evaluations were conducted in a sensory evaluation room at room temperature (25 °C). Each panelist received samples in their own booth. After achieving proficiency in (1) Training on TI analysis, the panelists proceeded to (2) Training on the beany flavor masking test.

(1) Training on TI analysis: The panelists were trained to evaluate changes in the flavor intensity of the flavoring materials over time using the TI method. The training utilized the TI program of the Fizz software (Version 2.0, Biosystems, Couternon, France). The panelists drank fake soymilk added to the flavoring materials and assessed the flavor TI. The panelists synchronized their breathing with a metronome set to 40 beats per minute (bpm), as flavor perception is known to be influenced by the rate of breathing [19,20,21]. During the training, the experimenters observed the upward and downward movement of the panelists’ shoulders from behind and checked for synchronization with a metronome. The training was completed when the panelists were able to produce smooth TI curves with high reproducibility.

(2) Training on the beany flavor masking test: The panelists were trained to evaluate the masking ability of the flavor materials against the beany flavor of soymilk using the 100-mm visual analog scale (VAS) method. To ensure a shared perception of the beany flavor in the soymilk among all panelists, they were provided with plain soymilk during the training and were briefed about the sensation of beany flavor. The panelists drank the plain soymilk to note their sense of the beany flavor as a control sample. The beany flavor masking ability of all 100 flavoring materials was assessed in our preliminary study. Pineapple essence (pineapple_e) was selected as a moderate sample for the panelist training, because it exhibited a medium-level beany flavor masking ability, reducing the beany intensity from 10 to around 7.5. The value of 7.5 represents a challenging evaluation point, as it is difficult to determine whether the beany flavor is completely masked (around 0) or not masked at all (around 10). This challenging material was intentionally used in the training process to assess the panel’s ability to accurately evaluate the beany flavor masking. The panelists tasted soymilk with pineapple_e or plain soymilk and rated the intensity of the beany flavor using a VAS sheet. The training was completed when the panelists evaluated and rated the intensity of the beany flavor of plain soymilk as around 10 and that of soymilk with added pineapple_e as around 7.5.

### 2.4. TI Analysis of the 100 Food-Flavoring Materials

The TI method is a time-series sensory analysis method that analyzes the temporal changes in the intensity of one sensory attribute [10,11]. It allows for a detailed analysis of the temporal changes in flavor intensity by providing numerical data on various TI parameters derived from the obtained TI curves [12,13]. In this study, the TI method was employed to evaluate the temporal changes in flavor intensity of the flavor materials. Specifically, each flavor material was added to the fake soymilk, and the trained panel evaluated the temporal changes in flavor intensity. The detailed methods are described below. All samples were consumed at room temperature throughout all evaluations. The panelists evaluated intensity changes in the flavor of the 100 flavoring materials using the TI analysis. To avoid interference from the beany flavor of soymilk, fake soymilk without beany flavor was used as the test medium. The time intensity program of the Fizz software was used for the analysis. The panelists were provided with the fake soymilk (10 mL) added to one of the flavoring materials (20 µL) just before the test. According to the instructions on the computer screen, the panelists kept the samples in their mouths for 5 s, which were then swallowed 10 s after the test started. The panelists controlled the cursor on a scale of 0 (no sense of flavor) to 100 (same intensity with reference samples) on the computer screen until the flavor sensation disappeared. The reference samples used were peppermint oil (peppermint_o) and rose oil (rose_o), which had strong flavor intensity and were given a score of 100 on the scale. These reference samples were presented to the panelists before the TI test, and the panelists memorized their maximum flavor intensity as 100. Subsequently, the panelists evaluated the flavor TI of each flavoring material. To confirm that the panelists were able to evaluate the flavor intensity of the materials relative to the reference samples, the TI data of white peach essence (white peach_e) and vanilla essence (vanilla_e), which had low flavor intensity in the preliminary study, were examined to see if their Imax values were approximately half compared to the reference samples. The TI data were collected every second. During the evaluation, the panelists synchronized their breathing with a metronome set to 40 beats per minute (bpm). Each material was evaluated five to eight times, with an average of three evaluations being used for analysis. The three representative curves were selected based on the intermediate measurements of three important TI parameters: maximum intensity of the curve (Imax), plateau start time (TsPl), and end time (Tend). This approach allowed for the exclusion of outlying data and ensured the reproducibility of the results.

The TI data were collected every second using the Fizz software, and 14 TI parameters were extracted from the obtained TI curves. These parameters included the following: start time (Tstart), end time (Tend), maximum intensity of the curve (Imax), plateau start time (TsPl), plateau end time (TePl), duration of the plateau phase (DurPl), duration of the increasing phase (DurInc), duration of the decreasing phase (DurDec), maximum slope measured in the increasing phase (SIMInc), maximum slope measured in the decreasing phase (SIMDec), the total area under the curve (AreaTse), area under the curve during the increasing phase (AreaInc), area under the curve during the decreasing phase (AreaDec), and area under the curve during the plateau phase (AreaPl).

### 2.5. Evaluation of the Masking Ability of the Food-Flavoring Materials against the Beany Flavor of Soymilk

The masking ability of the 100 flavor materials against the beany flavor of soymilk was evaluated using the VAS method. All samples were consumed at room temperature throughout the evaluation. The soymilk (10 mL) added to one of the food-flavoring materials (20 µL) was given to the panelists. Each flavor material was added to soymilk just before drinking. Each sample was labeled with a random three-digit number. As previously described in Section 2.4, the panelists breathed in synchrony with a metronome set at 40 bpm during the evaluation. The panel swallowed the plain soymilk to note their sense of the beany flavor as a control sample. During a one-minute break, the panelists rinsed their mouths with distilled water until the flavor sensation disappeared. Five seconds after the start of the evaluation, the panelists put the sample into their mouths, which was then swallowed 5 s later. The panel continued to evaluate the beany flavor until the flavor sensation disappeared. The panelists rated their sense of beany flavor using the VAS, ranging from “no sense of beany flavor” at the left end to “equivalent to the beany flavor of plain soymilk” at the right end. The median intensity of the beany flavor in the samples added with each flavoring material was calculated based on the VAS data. The masking ability of each flavoring material against the beany flavor was scored on a 10-point scale using the following equation: 10 − (median intensity of beany flavor) = masking score. For example, if the median intensity of the beany flavor in the sample with a flavoring material was 4.0, its masking score would be 6.0. To avoid panelist fatigue, each panelist evaluated five samples per test for a total of 20 tests (a total of 100 samples).

### 2.6. Statistical Analysis

The obtained TI data were analyzed using Fizz Calculations (version 2.61, Biosystemes). After smoothing the raw data (Figure 1A,B), an average curve was generated (Figure 1C). The values for the 14 TI parameters shown in Figure 1D were calculated from the average curve. The Mann–Whitney U test (two-tailed, α = 0.05) in GraphPad Prism 5 (version 5.03, GraphPad Software, CA, USA) was used to test for significant differences between the top 10 and bottom 10 masking materials (Figure 4). *p*-values of <0.05 were considered statistically significant. 

### 2.7. In Silico Analysis of the Time-Series Sensory Data for the Classification of the Food-Flavoring Materials

The obtained TI data were analyzed using unsupervised learning to classify the 100 flavoring materials based on their time-series profiles. The original Python script was used for the unsupervised learning. The data were first standardized to have a mean of 0 and a variance of 1 to avoid a strong influence from the parameters with large digits. Principal component analysis (PCA) was then performed to reduce the dimensionality of the data, and the calculated first and second principal components (PCA1 and PCA2) were used as the x- and y-axes, respectively. The number of clusters (k-value) was set to 3 because it resulted in a satisfactory separation of the data. Cluster analysis was performed using the k-means clustering algorithm from the scikit-learn library.

## 3. Results and Discussion

### 3.1. TI Analysis of the 100 Food-Flavoring Materials

To obtain fundamental sensory data for the 100 food-flavoring materials, the temporal changes in the flavor intensity of the materials were analyzed using the TI method. The TI curves were obtained for each flavor material, and then 14 parameter values were extracted from the curves (Appendix A). Four parameters, plateau start time (TsPl), plateau end time (TePl), end time (Tend), and the total area under the curve (AreaTse), were excluded from the data analysis since their values were similar to those of other parameters (Appendix A). As shown below, the values of some parameters were highly correlated. TsPl, TePl, and the duration of the increasing phase (DurInc) are parameters related to the increasing phase. The data values of these 3 parameters show a strong positive correlation with each other (correlation coefficient r = 0.99). Accordingly, DurInc was used for data analysis as the representative parameter related to the increasing phase. Four parameters, duration of the decreasing phase (DurDec), Tend, area under the curve during the decreasing phase (AreaDec), and AreaTse, are related to the decreasing phase. DurDec and Tend and AreaDec and AreaTse show high positive correlations, and their correlation coefficients were 0.99 and 0.98, respectively. Accordingly, DurDec and AreaDec were used for the data analysis as parameters related to the decreasing phase. Two parameters related to the plateau phase, the area under the curve during the plateau phase (AreaPl) and the duration of the plateau phase (DurPl), were also excluded from the analysis because none of the flavor materials reached a plateau phase (Appendix A). The obtained TI curves were divided into four areas, and eight parameters were selected to represent each area for data analysis: starting time (Tstart), maximum intensity (Imax), increasing phase (AreaInc, DurInc, and SIMInc), and decreasing phase (AreaDec, DurDec, and SIMDec). The values for those eight parameters were arranged in descending order from left to right (Figure 2A–H). The names and values of the materials for each TI parameter were listed in descending order in Appendix A. The values of four parameters, namely, AreaInc, DurInc, AreaDec, and DurDec, exhibited distinct trends based on the type of materials, essence, oil, and flavor (Figure 2C,D,F,G).

For a concise examination of the flavor materials’ characteristics based on each TI parameter, the top 10 and bottom 10 materials were selected for analysis. In terms of AreaInc and DurInc, 9 out of the top 10 materials (top 10% of the 100 materials) belong to the oil type (Table 3). Particularly, in regard to DurInc, 20 out of the top 25 materials were classified as oil type. These results indicate that the oil-type materials tended to have higher values of AreaInc and DurInc compared with the essence- and flavor-type materials. AreaInc represents the area under the curve of the increasing phase, while DurInc represents the duration of the increasing phase. Therefore, larger values of AreaInc and DurInc indicate a milder increase in flavor intensity in the increasing phase, indicating a slower flavor release in the increasing phase. Consequently, the oil-type materials were suggested to have a slower flavor release in the increasing phase compared with the essence- and flavor-type materials.

Regarding AreaDec and DurDec, 6 of the bottom 10 materials were essence-type materials, and for DurDec, 8 of the bottom 10 materials belonged to the essence type (Table 4). These results indicate that the essence-type materials tended to have lower values of AreaDec and DurDec compared with the oil- and flavor-type materials. AreaDec and DurDec represent the area under the curve during the decreasing phase and the duration of the decreasing phase, respectively. The smaller values of AreaDec and DurDec were associated with a sharper disappearance of flavor in the decreasing phase, indicating a lower flavor persistence in the decreasing phase. Therefore, the essence-type materials indicate lower flavor persistence in the decreasing phase compared with the oil- and flavor-type materials.

The differences in the time-series sensory profiles among oil, essence, and flavor-type materials may be attributed to their different hydrophobicity and volatility. Philippe et al. suggested that the hydrophobicity of flavor compounds affects the release of compounds from food model solutions [22]. The hydrophobic interaction between flavor compounds and proteins or lipids is an important molecular mechanism controlling flavor release [23,24]. In this study, the oil materials showed a higher lipophilicity and slower flavor release in the increasing phase compared with the essence- and flavor-type materials. The slower flavor release may be related to the hydrophobic interactions between the oil materials and proteins or lipids in the test solution.

Buffo et al. indicated that vapor pressure is an important parameter affecting flavor persistence as it is an index of substance volatility [25]. Moreover, they also showed an inverse correlation between vapor pressure and flavor persistence. Consistent with this report, our study also found that the essence types with higher volatility exhibited lower flavor persistence in the decreasing phase. The differences in the chemical properties of each material type are likely responsible for the differences in their TI profiles, as discussed above.

### 3.2. Analysis of the Relationship between TI Profile and Beany Flavor Masking Ability of the Food-Flavoring Materials

Since soy is an important alternative protein source due to its low resource consumption, masking its beany flavor is a significant challenge. In this study, soymilk was used as a model food with an unpleasant flavor. The masking ability of the 100 flavor materials against the beany flavor of soymilk was evaluated and scored on a 10-point scale; the higher the score, the higher the masking ability. The average masking score of the 100 flavoring materials was 7.1 (Figure 3, Appendix A). Cinnamon oil (cinnamon_o) and peppermint oil (peppermint_o) show the highest masking score, 9.9 points, while cocoa essential (cocoa_e) shows the lowest masking score, 1.2 points. Table 5 shows the flavor materials that have the top 10 and bottom 10 masking scores. Seven of the top 10 materials were classified as oil types as follows: cinnamon_o, peppermint_o, orange essential_o, pineapple_o, rose_o, yuzu essential_o, and cheese_o. The bottom 10 materials were composed of five flavor-type materials (kabosu_f, sudachi_f, lemon_f, kyoho grape_f, and framboise_f), four essence-type materials (cocoa_e, white peach_e, lemon_e, and apple_e), and two oil-type materials (raspberry_o and apple_o).

To extract the important TI parameters related to the beany flavor masking, the values of the eight TI parameters were compared between the top 10 and bottom 10 materials (Figure 4). The values of five parameters, Tstart, Imax, DurInc, SIMInc, and SIMDec, were not significantly different between the top 10 and bottom 10 materials, whereas the values of three parameters, AreaInc, AreaDec, and DurDec, were significantly higher in the top 10 group than those in the bottom 10 group. These results suggested that these three TI parameters are important for the beany flavor masking by the flavor materials; an effective masking can be achieved by focusing on the TI profiles of flavoring materials.

### 3.3. Classification of the Food-Flavoring Materials by Unsupervised Machine Learning Based on Time-Series Sensory Data

Machine learning is one of the powerful data analysis approaches that can derive rules from complex datasets. To analyze the relationships between the sensory profiles and the beany flavor masking ability of the 100 flavor materials, the materials were classified based on their sensory profiles. Their TI profile datasets were subjected to PCA and cluster analysis using unsupervised machine learning. The flavor materials were divided into three clusters: clusters 0, 1, and 2 (Figure 5). Cluster 2 was clearly separated from clusters 0 and 1. The contribution of PCA1 and PCA2 to the clustering analysis was 98% and 2%, respectively. This means that the contribution of PCA2 was very small, while PCA1 accounted for the majority of the clustering. Accordingly, the flavor materials can be classified by the PCA1 components. Table 6 and Appendix A show the sample name and masking score of the materials that compose cluster 2 and clusters 0 and 1, respectively. The average masking score of the 8 materials was 8.2 points, which was significantly higher than that of clusters 0 and 1, i.e., 6.9 points (Mann–Whitney U test, two-tailed, α = 0.05, *p* < 0.05). The 8 materials that compose cluster 2 contain 2 of the top 10 materials, but not the bottom 10 materials. Accordingly, the flavor materials can be classified by the PCA1 components. Moreover, the high values of PCA1 can be associated with the high masking ability of the materials. PCA1 was composed of the eight TI parameters: Imax, Tstart, DurInc, DurDec, SIMInc, SIMDec, AreaInc, and AreaDec. Since the data were standardized in the initial stages of the analysis, the coefficients of the explanatory variables can be used for the quantitative comparison of their contributions. Among the eight explanatory variables, the coefficients of AreaDec and AreaInc were higher than those of the others, indicating that they had a greater influence on PCA1 (Table 7). Therefore, it can be inferred that the components of PCA1, specifically AreaDec and AreaInc, are important for the classification and the high masking ability of the flavoring materials. AreaInc and AreaDec are parameters related to the flavor release in the increasing phase and that in the decreasing phase, respectively. Because the coefficient of AreaDec was approximately six times larger than that of AreaInc for PCA1, the flavor release in the decreasing phase is a more important factor than that in the increasing phase for the classification and the high masking ability of the flavor materials. As a result of the PCA and the cluster analysis using unsupervised machine learning, the flavor materials were classified according to their TI profiles. Moreover, TI parameters related to the classification and the beany flavor masking ability were extracted for the flavor materials. The application of machine learning to TI data analysis was considered a valuable approach capable of deriving complex rules from large datasets.

AreaInc and AreaDec, particularly AreaDec, were identified as important TI parameters that are related to the beany flavor masking in machine learning analysis and individual analysis. The in silico analysis successfully determined the contribution ratio of each TI parameter to the classification of the flavor materials, which is related to their masking ability. Machine learning derives certain rules from complex phenomena. In our previous studies, the rules of ligands on proton-coupled oligopeptide transporters and transient receptor potential (TRP) channels were clarified [26,27]. This study showed the rules for masking by food-flavoring materials.

AreaDec, a TI parameter related to the flavor release in the decreasing phase, was suggested to be important for the beany flavor masking by flavor materials. Effective masking can be achieved by focusing on the flavor material’s AreaDec. In the present study, the AreaDec value of soymilk was 2074, indicating a strong flavor intensity in the decreasing phase. The mean value of AreaDec for masking the top 10 materials was 1763, which was close to that of soymilk, suggesting that the top 10 materials effectively masked the beany flavor of soymilk in the decreasing phase. Numerous studies have focused on controlling the flavor release from foods, e.g., glycerol, polysaccharides, and protein addition, and flavor encapsulation is one of the approaches used to modify the release of flavor compounds [28,29,30,31]. Jones et al. reported that the addition of glycerol and polysaccharides enhanced and weakened the flavor intensity of white wine, respectively [28]. Santos et al. showed that the encapsulation of mint flavor using gum arabic increased the persistence of the flavor in chewing gum [32]. Accordingly, controlling the release of flavor materials would enable more effective masking.

## 4. Conclusions

To provide a new perspective on the development of masking agents, this study examined the application of the time-series sensory profiles of food-flavoring materials. The individual analysis between the TI parameters and the masking ability indicated that the values of AreaInc, DurDec, and AreaDec were significantly higher in the top 10 materials with high masking scores compared to the bottom 10 materials. Machine learning-based PCA and cluster analysis of the flavoring materials presented AreaInc and AreaDec as TI parameters contributing to the classification of flavor materials and their masking ability. Both the individual analysis and the in silico analysis consistently indicate the importance of AreaDec, a TI parameter associated with flavor release during the decreasing phase, in the masking of beany flavor by flavoring materials. Time-series sensory evaluation is a useful method for understanding the sensory characteristics of foods. In addition to individual analysis, machine learning, which can derive complex rules from large amounts of data, will be a powerful tool for developing masking agents. In the future, an approach combining time-series sensory profiles and machine learning will lead to a more strategic masking process.

## Figures and Tables

**Figure 1 foods-12-02752-f001:**
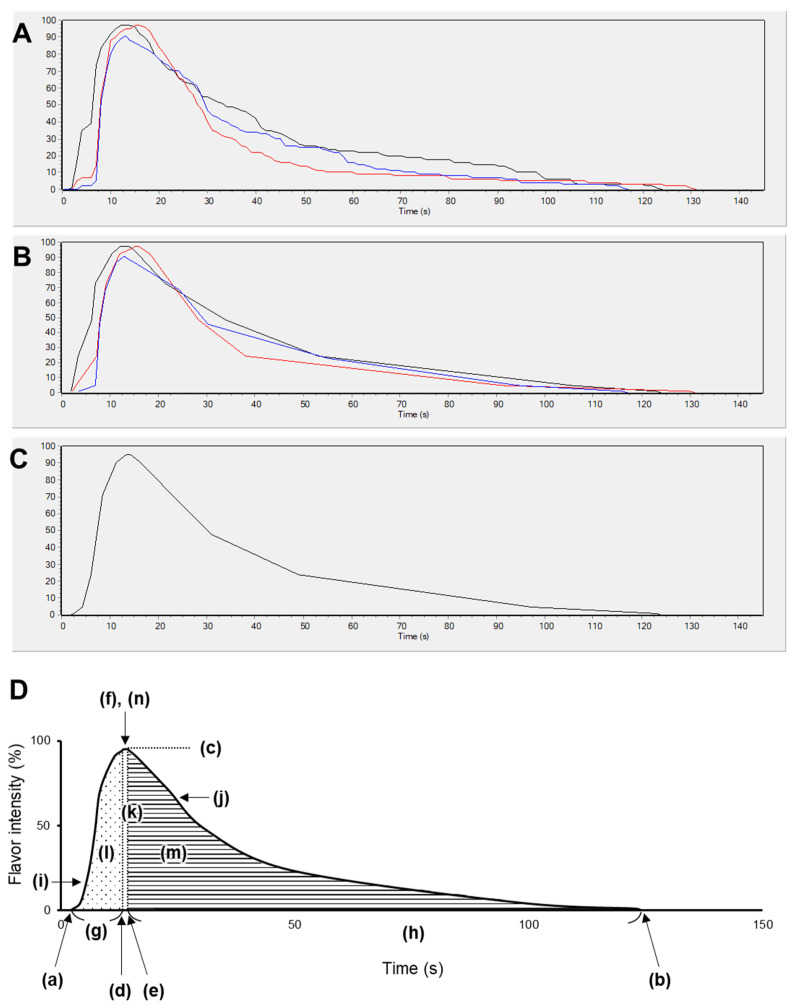
Example of time-intensity (TI) curve analysis and 14 TI parameters. Maple flavor (Maple_f) is shown as an example. (**A**) Raw data, (**B**) data after smoothing, (**C**) average curve, and (**D**) 14 TI parameters extracted from the average TI curve: (a) start time (Tstart), (b) end time (Tend), (c) maximum intensity of the curve (Imax), (d) plateau start time (TsPl), (e) plateau end time (TePl), (f) duration of the plateau phase (DurPl), (g) duration of the increasing phase (DurInc), (h) duration of the decreasing phase (DurDec), (i) maximum slope measured in the increasing phase (SIMInc), (j) maximum slope measured in the decreasing phase (SIMDec), (k) total area under the curve (AreaTse), (l) area under the curve during the increasing phase (AreaInc), (m) area under the curve during the decreasing phase (AreaDec), and (n) area under the curve during the plateau phase (AreaPl).

**Figure 2 foods-12-02752-f002:**
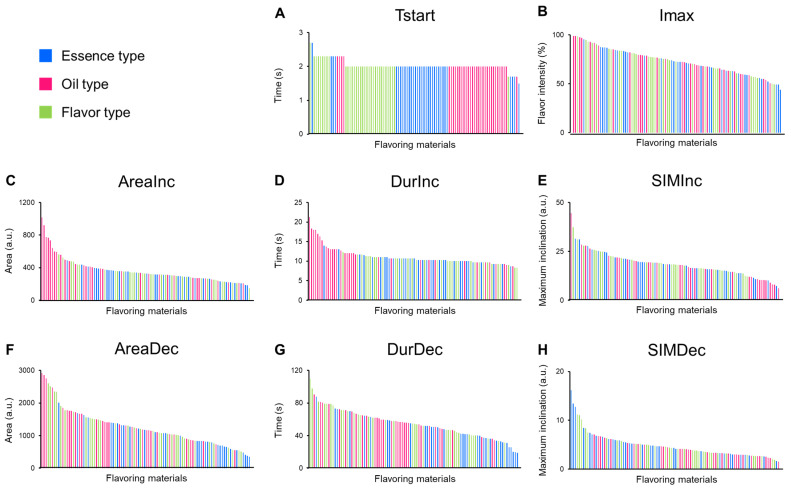
Comparison of 8 time-intensity (TI) parameter values among the 100 food-flavoring materials. The data values of the 100 flavor materials were arranged in descending order from left to right. The eight TI parameters are as follows: (**A**) Tstart, (**B**) Imax, (**C**) AreaInc, (**D**) DurInc, (**E**) SIMInc, (**F**) AreaDec, (**G**) DurDec, and (**H**) SIMDec. The colors of the bars indicate the material types: blue for essence type, magenta for oil type, and light green for flavor type. a.u.: arbitrary unit.

**Figure 3 foods-12-02752-f003:**
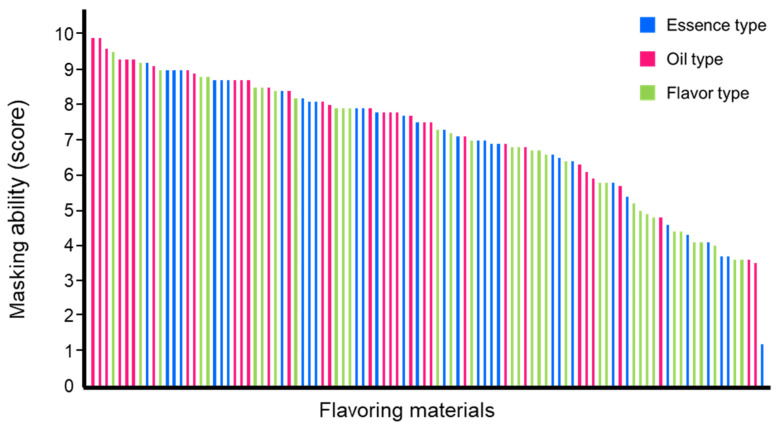
The masking score of the 100 flavor materials against beany flavor. The masking ability (score) of the 100 flavoring materials against the beany flavor of soymilk was arranged in descending order from left to right. The masking score of each flavoring material was calculated on a 10−point scale using the following equation: 10 − (median intensity of beany flavor) = masking score. The color of the bar indicates the material types: blue for essence type, magenta for oil type, and light green for flavor type.

**Figure 4 foods-12-02752-f004:**
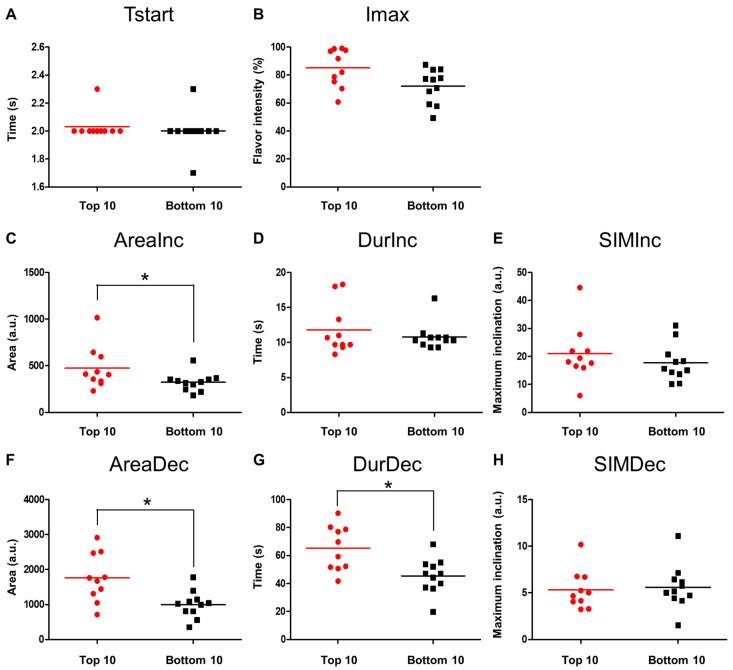
Comparison of TI parameter values between the top 10 and bottom 10 masking score materials. The eight TI parameter values, (**A**) Tstart, (**B**) Imax, (**C**) AreaInc, (**D**) DurInc, (**E**) SIMInc, (**F**) AreaDec, (**G**) DurDec, and (**H**) SIMDec were compared between the top 10 (red) and bottom 10 (black) masking score materials. The Mann–Whitney U test (two-tailed, α = 0.05) was used to test the significance of the differences between the two groups. * indicates *p* < 0.05.

**Figure 5 foods-12-02752-f005:**
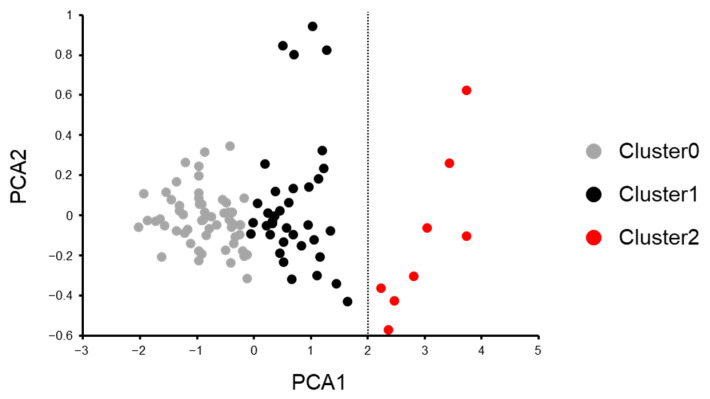
Classification of the 100 flavoring materials based on their TI profiles: principal component analysis and cluster analysis using unsupervised learning. Cluster analysis using the k-means method (k = 3) was performed for the flavor materials based on their TI profiles, with the first principal component (PCA1) and the second principal component (PCA2) set as the x- and y-axes, respectively. Gray, cluster 0; black, cluster 1; and red, cluster 2.

**Table 1 foods-12-02752-t001:** Classification of the flavoring materials used in this study.

Material Type	Notation	Main Solvents	Chemical Features
Essence type	e	Alcohols	Water-soluble/High volatility
Oil type	o	Glycerol fatty acid esters/Vegetable oils	Oil-soluble/Low volatility
Flavor type	f	Propylene glycol/Glycerin	Intermediate polarity/High volatility

The flavor materials were classified into three types based on their chemical features: essence type (e), oil type (o), and flavor type (f). Each material name was shown as flavor name_type (e, o, or f) (e.g., orange oil and maple flavor were noted as orange_o and maple_f, respectively). The type (e, o, or f) of the 100 flavoring materials is shown in Appendix A.

**Table 2 foods-12-02752-t002:** Composition of the fake soymilk.

Ingredient	g
Almond milk	34
Soy protein RT-1	3.0
Dextrin	3.0
Water	60

**Table 3 foods-12-02752-t003:** Top 10 AreaInc and DurInc materials.

Top	AreaInc	Area (a.u.)	DurInc	Time (s)
1	Orange essential_o	1016	Satsuma mandarin essential_o	21.3
2	Satsuma mandarin essential_o	919	Yuzu essential_o	18.3
3	Coconut_o	776	Coconut_o	18.0
4	Orange_o	765	Orange essential_o	18.0
5	Cherry_o	735	Orange_o	17.0
6	Yuzu essential_o	644	Apple_o	16.3
7	Rose_o	597	Banana_o	15.3
8	Mango_o	595	Orange_e	14.0
9	Maple_f	563	Cherry_o	13.7
10	Apple_o	558	Rose_o	13.3

The name of the flavoring materials was denoted as flavor name_type (e, o, or f). For example, orange essence, cherry oil, and maple flavor were shown as orange_e, cherry_o, and maple_f, respectively.

**Table 4 foods-12-02752-t004:** Bottom 10 AreaDec and DurDec materials.

Bottom	AreaDec	Area (a.u.)	DurDec	Time (s)
1	White peach_e	352	Pineapple_e	18.7
2	Pineapple_e	380	White peach_e	19.7
3	Honey_e	414	Cheese_e	20.0
4	Caramel_e	484	Caramel_e	25.7
5	White peach_f	515	Cranberry_e	25.7
6	Honey_f	539	Orange_e	31.0
7	Cheese_e	552	Pineapple_f	31.3
8	Raspberry_o	554	Cinnamon_e	31.7
9	Brown sugar_f	565	Honey_e	33.0
10	Orange_e	594	White peach_f	33.3

**Table 5 foods-12-02752-t005:** Top 10 and bottom 10 masking score materials.

Top	Material Name	Masking Score	Bottom	Material Name	Masking Score
1	Cinnamon_o	9.9	1	Cocoa_e	1.2
2	Peppermint_o	9.9	2	Raspberry_o	3.5
3	Orange essential_o	9.6	3	Apple_o	3.6
4	Herb_f	9.5	4	Kabosu_f	3.6
5	Pineapple_o	9.3	5	Sudachi_f	3.6
6	Rose_o	9.3	6	White peach_e	3.7
7	Yuzu essential_o	9.3	7	Lemon_e	3.7
8	Coffee_f	9.2	8	Lemon_f	4.0
9	Rose_e	9.2	9	Apple_e	4.1
10	Cheese_o	9.1	10	Kyoho grape_f	4.1
			11	Framboise_f	4.1

**Table 6 foods-12-02752-t006:** Eight materials belonging to cluster 2.

Material Name	Masking Score
Satsuma mandarin essential_o	8.4
Rose_o	9.3
Cherry_o	9.0
Maple_f	7.0
Herb_f	9.5
Plum_f	8.8
Custard_f	4.8
Banana_e	9.0
Average	8.2

**Table 7 foods-12-02752-t007:** Components of principal component analysis 1.

Imax	Tstart	DurInc	DurDec	SIMInc	SIMDec	AreaInc	AreaDec
0.015	−6.00 × 10^−5^	0.0015	0.026	0	0	0.167	0.985

## Data Availability

The data underlying this article are available in the article and in its online Appendix A.

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
