# Peer review of "Time-Series Sensory Analysis Provided Important TI Parameters for Masking the Beany Flavor of Soymilk"

_foods, 2023, doi:10.3390/foods12142752_

Round 1

Reviewer 1 Report

1. Tue study is very interesting. However, there are some important points that present difficult interpretations to the reader.

2. Table 2, mentioned as supplementary material, should be included in the main manuscript. The composition of the "fake soymilk without beany flavor" is important information for the reader to understand the research.

3. The conclusion needs to be more clear and objective.

4. Topic 2.3. Panelists and training

Please, explain how the training was conducted in steps 1) Training on TI analysis and 2) Training on beany flavor masking test.

5. Were references used to form sensory memory? How were they used? And how was the training validated? Understanding these steps is crucial for comprehending the article.

6. Are there no repetitions in the TI tests conducted by the panelists?

7. It would be highly important to have repetitions of panelists to T-I test for each one of the stimuli in order to confirm the training and repeatability power, for a more robust interpretation of the data.

8. The key-word "Consumer" don't is appropriate, maybe panelists or assessors

The Quality of English Language is good. But minor editing of English language required.

Reviewer 2 Report

Overall the manuscpit is novel and meaningful, but I have some quesitons which needs the authors to clairify:

I doubt whether the “fake soymilk” is reasonable control, since the composition of the fake soymilk include almond milk. Even though almond milk does not have beany flavor, the almond has unique flavor which may influence the accuracy of the sensory results.

The pineapple did mask the beany flavor, however, whether the panelist could correctly identify the beany flavor without the pineapple essence was a question.

The panelists tastes 100 materials, how to prevent the fatigue?

Reviewer 3 Report

All tables are missing?

How do the authors counter the halo dumping effect from TI - as it is well known to have that.

What exactly did the author mean here by drawing TI curve? and how can the author conclude high reproducibility? More info needed.

It really confuses me on how the calibration was done here, how can the author decide that 10 and 7.5 is sufficient? Shouldn't there be 2 reference points for the intensity in calibration?

The metronome is great but how can the author really check their breathing is actually synced?

Why the number of cluster is set at 3? If it's unsupervised shouldn't we let the script decide the number of cluster?

The removal of the parameters should be further justified. What do u mean exactly by similar? High correlation?

Fig 2 are nice but doesn't tell me much. So what does each bar represents individually? Is it possible to actually put the name of the flavours there?

The selection of top and btm 10% needs to be further justified. Of course there'll be significant difference (or not) in some TI parameters, but I don't quite get the aim here?

So what are the cluster results? Did it succesfully somehow cluster or extract important features from the 100 flavourings? 

How did the author conclude that AreaInc and Dec is the most improtant one based on the cluster analysis? Quite lost on the claim here.

Table 1. is a nice legend to have but becomes irrelevant as the authors do not show the sample names (except for top and bottom samples?)   Table 2-5. Still unclear to me why top and bottom 10 is selected.   Table 6. I don't quite understand why only cluster 2 results is  shown here, additionally it seems that the clustering was done poorly where you can see that there is one that is out of range from the average (e.g., custard?)

Table 7, I'm quite confused here, why only PC1 is shown? are these factor scores?

English reads fine

Reviewer 4 Report

The manuscript addresses a study on identifying potential flavorings to mask unwanted flavors. The relevance is in the number of evaluated materials (100). However, authors should carefully reread the instructions for authors for formatting the manuscript. Tables are not present in the manuscript text. The lines are not numbered, ...

Authors must explain the reason for the criteria established to complete the training. Why is the score difference (10 versus 7.5) significant for completing the training?

Why only use three evaluations if the materials were evaluated five to eight times?

Round 2

Reviewer 1 Report

The authors improved the manuscript. I think it can be accepted with the changes made.

Author Response

Reviewer 1

We sincerely appreciate the time and effort you dedicated to reviewing our manuscript.

We apologize for the confusion, but we could not locate specific comments regarding the revisions from Reviewer 1. However, we received additional questions from Reviewer 3 regarding the description of the methods, which prompted further revisions. We have incorporated the comments from Reviewer 3 and made corresponding modifications in the manuscript. Please find the comments from Reviewer 3 and our responses below.

#2, OK so how does the TI look like on the fake soymilk? The suggestion "would be minimal" is not a sufficient justification here.

Response: In our preliminary study, we attempted to evaluate the flavor TI of the fake soymilk. However, due to the fact that the fake soymilk was nearly tasteless and odorless, it proved challenging to assess the TI accurately.

#3, Checking upward and downward movement of shoulder suggests breathing movement not preformance or calibration of a stimulus. So elaborate a bit more here, the flavoring was evaluated 5-8x and only 3 were selected and then averaged? If yes, then how did the author choose the 3?

Response: Observation of the upward and downward movement of body parts, such as the chest and shoulders, is the most common and non-invasive method to accurately check the breathing tempo. Analyzing exhaled gas is a more rigorous approach, but it requires covering the panelists' mouth with a device during the tests, which is not suitable for the sensory evaluation. In this study, the observation of panelists' shoulder movement to assess breathing tempo is considered an appropriate approach.

Each flavoring material was evaluated multiple times (5 to 8 times), and the average of three evaluations was used for the analysis. The three representative curves were selected based on the intermediate measurements of three important TI parameters: Imax, TsPl, and Tend. We have added the relevant description to the manuscript (lines 185 to 187).

#4, So pineapple_e is selected as the lowest point, how did the author decide on this? Did the author assessed all the aromas?

Response: The beany flavor masking ability of all 100 flavoring materials was assessed in our preliminary study. Among the materials, pineapple_essence (pineapple_e) exhibited a medium-level beany flavor masking ability, reducing the beany intensity from 10 to approximately 7.5. The value of 7.5 represents a challenging evaluation point, as it is difficult to determine whether the beany flavor is completely masked (around 0) or not masked at all (around 10). This challenging material was intentionally used in the training process to assess the panel's ability to accurately evaluate the beany flavor masking. It should be noted that pineapple_e was not selected as the lowest point sample. We have added the relevant sentences to the manuscript (lines 148 to 149). Please refer to the manuscript (lines 149 to 156).

#5, Standing behind the participants evokes lots of biases and problems - see participants/response bias, which can also evoke social desirability effect?

Response: The experimenters observed the upward and downward movement of the panelists' shoulders from behind and checked for synchronization with a metronome during the training. It should be noted that these observations were performed during the training and did not have any impact on the results obtained. Please refer to the manuscript (lines 185 to 187).

#6, Check some index for cluster stabilities (e.g., Hartigan), this can be useful to support this claim.

Response: The calculation of indices related to cluster stability, such as the Hartigan score and adjusted Rand score, typically requires the identification of true positives. However, in this study using sensory evaluation data, it is not possible to assign true positives. Specifically, it is not feasible to assign specific flavor materials as definitive references for each cluster (clusters 0, 1, and 2). Given the inability to define true positives, it is challenging to assess the validity of clustering using indices related to cluster stability.

Starting from #9, the line numbers are incorrect? But the mansucript reads better now. Needing further clarifcation on the methods and addressing biases

Response: Please refer to the attached PDF file. We have verified that the line numbers match with the PDF. Please note that line numbers may vary depending on individual settings of the proofreading function in Word.

Reviewer 3 Report

#2, OK so how does the TI look like on the fake soymilk? The suggestion "would be minimal" is not a sufficient justification here.

#3, Checking upward and downward movement of shoulder suggests breathing movement not preformance or calibration of a stimulus. So elaborate a bit more here, the flavoring was evaluated 5-8x and only 3 were selected and then averaged? If yes, then how did the author choose the 3?

#4, So pineapple_e is selected as the lowest point, how did the author decide on this? Did the author assessed all the aromas?

#5, Standing behind the participants evokes lots of biases and problems - see participants/response bias, which can also evoke social desirability effect?

#6, Check some index for cluster stabilities (e.g., Hartigan), this can be useful to support this claim

Starting from #9, the line numbers are incorrect? But the mansucript reads better now. Needing further clarifcation on the methods and addressing biases

English reads fine

Author Response

Reviewer 3

We sincerely appreciate the time and effort you dedicated to reviewing our paper. Your thoughtful comments and suggestions have been immensely helpful in strengthening our research.

#2, OK so how does the TI look like on the fake soymilk? The suggestion "would be minimal" is not a sufficient justification here.

Response: In our preliminary study, we attempted to evaluate the flavor TI of the fake soymilk. However, due to the fact that the fake soymilk was nearly tasteless and odorless, it proved challenging to assess the TI accurately.

#3, Checking upward and downward movement of shoulder suggests breathing movement not preformance or calibration of a stimulus. So elaborate a bit more here, the flavoring was evaluated 5-8x and only 3 were selected and then averaged? If yes, then how did the author choose the 3?

Response: Observation of the upward and downward movement of body parts, such as the chest and shoulders, is the most common and non-invasive method to accurately check the breathing tempo. Analyzing exhaled gas is a more rigorous approach, but it requires covering the panelists' mouth with a device during the tests, which is not suitable for the sensory evaluation. In this study, the observation of panelists' shoulder movement to assess breathing tempo is considered an appropriate approach.

Each flavoring material was evaluated multiple times (5 to 8 times), and the average of three evaluations was used for the analysis. The three representative curves were selected based on the intermediate measurements of three important TI parameters: Imax, TsPl, and Tend. We have added the relevant description to the manuscript (lines 185 to 187).

#4, So pineapple_e is selected as the lowest point, how did the author decide on this? Did the author assessed all the aromas?

Response: The beany flavor masking ability of all 100 flavoring materials was assessed in our preliminary study. Among the materials, pineapple_essence (pineapple_e) exhibited a medium-level beany flavor masking ability, reducing the beany intensity from 10 to approximately 7.5. The value of 7.5 represents a challenging evaluation point, as it is difficult to determine whether the beany flavor is completely masked (around 0) or not masked at all (around 10). This challenging material was intentionally used in the training process to assess the panel's ability to accurately evaluate the beany flavor masking. It should be noted that pineapple_e was not selected as the lowest point sample. We have added the relevant sentences to the manuscript (lines 148 to 149). Please refer to the manuscript (lines 149 to 156).

#5, Standing behind the participants evokes lots of biases and problems - see participants/response bias, which can also evoke social desirability effect?

Response: The experimenters observed the upward and downward movement of the panelists' shoulders from behind and checked for synchronization with a metronome during the training. It should be noted that these observations were performed during the training and did not have any impact on the results obtained. Please refer to the manuscript (lines 185 to 187).

#6, Check some index for cluster stabilities (e.g., Hartigan), this can be useful to support this claim.

Response: The calculation of indices related to cluster stability, such as the Hartigan score and adjusted Rand score, typically requires the identification of true positives. However, in this study using sensory evaluation data, it is not possible to assign true positives. Specifically, it is not feasible to assign specific flavor materials as definitive references for each cluster (clusters 0, 1, and 2). Given the inability to define true positives, it is challenging to assess the validity of clustering using indices related to cluster stability.

Starting from #9, the line numbers are incorrect? But the mansucript reads better now. Needing further clarifcation on the methods and addressing biases

Response: Please refer to the attached PDF file. We have verified that the line numbers match with the PDF. Please note that line numbers may vary depending on individual settings of the proofreading function in Word.
